# Identification of Novel Cyclooxygenase-1 Selective Inhibitors of Thiadiazole-Based Scaffold as Potent Anti-Inflammatory Agents with Safety Gastric and Cytotoxic Profile

**DOI:** 10.3390/molecules28083416

**Published:** 2023-04-12

**Authors:** Michelyne Haroun, Maria Fesatidou, Anthi Petrou, Christophe Tratrat, Panagiotis Zagaliotis, Antonis Gavalas, Katharigatta N. Venugopala, Hafedh Kochkar, Promise M. Emeka, Nancy S. Younis, Dalia Ahmed Elmaghraby, Mervt M. Almostafa, Muhammad Shahzad Chohan, Ioannis S. Vizirianakis, Aliki Papadimitriou-Tsantarliotou, Athina Geronikaki

**Affiliations:** 1Department of Pharmaceutical Sciences, College of Clinical Pharmacy, King Faisal University, Al-Ahsa 31982, Saudi Arabia; 2School of Pharmacy, Aristotle, University of Thessaloniki, 54124 Thessaloniki, Greece; 3Division of Infectious Diseases, Weill Cornell Medicine, New York, NY 10065, USA; 4Department of Biotechnology and Food Technology, Faculty of Applied Sciences, Durban University of Technology, Durban 4001, South Africa; 5Department of Chemistry, College of Science, Imam Abdulrahman Bin Faisal University, Dammam 31441, Saudi Arabia; 6Basic & Applied Scientific Research Center, Imam Abdulrahman Bin Faisal University, Dammam 31441, Saudi Arabia; 7Department of Pharmacy Practice, College of Clinical Pharmacy, King Faisal University, Al-Ahsa 31982, Saudi Arabia; 8Department of Chemistry, College of Science, King Faisal University, Al-Ahsa 31982, Saudi Arabia; 9Biomedical Sciences Department, College of Clinical Pharmacy, King Faisal University, Al-Ahsa 31982, Saudi Arabia; 10Laboratory of Pharmacology, School of Pharmacy, Aristotle University of Thessaloniki, 54124 Thessaloniki, Greece; 11Department of Health Sciences, School of Life and Health Sciences, University of Nicosia, 2417 Nicosia, Cyprus

**Keywords:** molecular modeling, thiadiazole, enzyme inhibition, anti-inflammatory, ulcerogenic effect, cyclooxygenase, lipoxygenase, cytotoxicity

## Abstract

Major obstacles faced by the use of nonsteroidal anti-inflammatory drugs (NSAID) are their gastrointestinal toxicity induced by non-selective inhibition of both cyclooxygenases (COX) 1 and 2 and their cardiotoxicity associated with a certain class of COX-2 selective inhibitors. Recent studies have demonstrated that selective COX-1 and COX-2 inhibition generates compounds with no gastric damage. The aim of the current study is to develop novel anti-inflammatory agents with a better gastric profile. In our previous paper, we investigated the anti-inflammatory activity of 4-methylthiazole-based thiazolidinones. Thus, based on these observations, herein we report the evaluation of anti-inflammatory activity, drug action, ulcerogenicity and cytotoxicity of a series of 5-adamantylthiadiazole-based thiazolidinone derivatives. The in vivo anti-inflammatory activity revealed that the compounds possessed moderate to excellent anti-inflammatory activity. Four compounds **3**, **4**, **10** and **11** showed highest potency (62.0, 66.7, 55.8 and 60.0%, respectively), which was higher than the control drug indomethacin (47.0%). To determine their possible mode of action, the enzymatic assay was conducted against COX-1, COX-2 and LOX. The biological results demonstrated that these compounds are effective COX-1 inhibitors. Thus, the IC_50_ values of the three most active compounds **3**, **4** and **14** as COX-1 inhibitors were 1.08, 1.12 and 9.62 μΜ, respectively, compared to ibuprofen (12.7 μΜ) and naproxen (40.10 μΜ) used as control drugs. Moreover, the ulcerogenic effect of the best compounds **3**, **4** and **14** were evaluated and revealed that no gastric damage was observed. Furthermore, compounds were found to be nontoxic. A molecular modeling study provided molecular insight to rationalize the COX selectivity. In summary, we discovered a novel class of selective COX-1 inhibitors that could be effectively used as potential anti-inflammatory agents.

## 1. Introduction

Nonsteroidal anti-inflammatory drugs (NSAIDs) are a chemically heterogeneous medication family of drugs approved by the FDA due to their analgesic and anti-inflammatory efficacy [1]. Their mode of action is based mainly on cyclooxygenase (COX) enzyme inhibition. COX is a key enzyme in the formation of prostanoids, existing in two isoforms: an inducible enzyme cyclooxygenase (COX)-2) and a constitutive from COX-1 [2]. COX inhibitors are widely prescribed medications in the management of chronic inflammatory conditions [3]. Conversely, common NSAIDS are implicated with a multitude of undesired effects including headache, vomiting/nausea, diarrhea, constipation, abdominal pain, allergies and drowsiness. Heart failure, hypertensive heart disease and heart attacks can also be correlated with their treatment. The most notable adverse effects of COX-1 inhibitors in patients are bleeding and gastrointestinal (GI) ulceration, but additionally kidney failure [4,5], while the essential increased risks of myocardial infarction and cardiovascular thrombotic events are associated with selective COX-2 inhibitors [3]. Administration of some COX-2 inhibitors has been accompanied with their withdrawal from the market, as in the case of the drugs valdecoxib and rofecoxib, caused by the appearance of acute cardiovascular incidents [6,7,8].

GI toxicity was mainly observed with the use of non-selective COX-1/-2 inhibitors of traditional NSAIDs, and COX-1 has been, for a long time, wrongly considered to be responsible for the gastric side effects. In consequence, the research area on the development of potent anti-inflammatory/analgesic agents has been focused on the discovery of a novel class of COX-2 selective inhibitors, aiming at preventing the undesired gastric side effects. Two decades ago, the study of Tanaka et al. demonstrated with the use of SC-560 and celecoxib as selective COX-1 and COX-2 inhibitors, respectively, that gastric ulceration is caused when both COX-1 and COX-2 are inhibited, and the supporting evidence demonstrated that the selective COX-1 inhibition failed to generate gastric lesions [9]. Much earlier, Wallace et al. established that inhibition of both COX-1 and COX-2 is needed for NSAID-induced gastric injury, and stomach irritation was not observed when COX-1 was selectively inhibited [10]. Langenbach et al. revealed that COX-1 knockout mice did not spontaneously evolve gastric lesion, as further evidence that COX-1 inhibition alone is insufficient to produce gastrointestinal toxicity [11]. Later, Tanaka et al. [12] reported the upregulation of cyclooxygenase-2 expression in gastric mucosa, mediated by COX-1 selective inhibition, and resulting in increased intestinal prostaglandin E2 production, which represents the major contributor to mucosal integrity maintenance. Nonetheless, the exact mechanism of upregulation COX-2 expression, when COX-1 is solely inhibited, remains yet to be clarified. 

Only a small number of selective cyclooxygenase-1 inhibitors were developed as healthful anti-platelets and anti-analgesic agents with improved gastric profile (Figure 1) [13,14]. 

Among them, Mofezolac, a diarylisoxazole, is a unique approved medicine in Japan as Disopain for its potent analgesic properties [15]. The ulcer indices of Mofezolac and Indomethacin were reported to display 0.98 and 10.29, respectively, with a dose of 10 mg, indicating ulcer liability improvement [16]. Similarly, diarylthiazole [17] and (S)-F-ibuprofen [18] were both reported to display low gastric damage. However, the following COX-1 selective inhibitors, TFAP [19], P6 [20], SC-560 [9], FR122047 [21] and *N*-methylpiperidinylindole [22], were found to be devoid of any gastric toxicity.

Regrettably, COX-1 remains poorly investigated as a molecular target for the treatment of other maladies as compared to COX-2. Few studies have revealed involvement of the COX-1 enzyme in many disorder diseases comprising thrombosis [23], neuro-inflammation [15,24], and cancers [25]. For example, several ovarian tumor cells have indicated cyclooxygenase-1 enzyme heightened expression devoid of any implication of cyclooxygenase-2 expression [26]. Other reports clearly manifested that COX-1 selective inhibitor demonstrated high efficiency toward cyclooxygenase-1 expression of human ovarian adenocarcinoma cells, while celecoxib, a cyclooxygenase-2-selective inhibitor, was inefficacious in either decreasing the biosynthesis of prostaglandin or tumor growth and invasion [27]. Mofezolac was additionally reported to demonstrate in vivo anticancer potency against colon tumors [28]. SC-560 selectively inhibited the proliferation of colon cancer cells by blocking transition from the G0/G1 to the S phase of the cell cycle [29,30]. Additionally, cyclooxygenase-1 may be critically beneficial in cardiac protection with reperfusion in acute myocardial infarction [31]. Therefore, it is of urgent significance to develop novel anti-inflammatory agents with efficacious curative effects and reduced side effects as substitutes of traditional anti-inflammatory drugs [8].

Based on the above, COX-1 appears to provide numerous therapeutic advantages in the development of new potent derivatives as anticancer, antiplatelet and anti-inflammatory agents that are capable of reducing gastric toxicity.

Recently, we reported the anti-inflammatory activity of a series of 5-methylthiazole-thiazolidinone derivatives identified as COX-1 selective inhibitors (Figure 2) [32]. We also described the biological activity of 5-adamantylthiadiazole-thiazolidinone conjugates as potent antimicrobial agents (Figure 2) [33]. With these compounds in hand and inspired by the anti-inflammatory property of 5-methylthiazole-thiazolidinones, it was conceivable to investigate the impact of the thiadiazole ring linked to thiazolidinone core and the replacement of the 5-methyl group in thiazole by the 5-adamantyl group in thiadiazole on the anti-inflammatory activity. The rationale of the designed molecules was to revert the COX selectivity in favor to COX-2 by exploiting the size difference between the COX-2/COX-1 active sites. It appeared to us that the adamantyl group would be the ideal substituent to achieve this objective. 

In a continuation of developing novel antimicrobial agents [34,35,36,37,38,39,40] and anti-inflammatory agents [41,42,43,44], we wish to disclose the biological activity of a series of thiadiazole–thiazolidinone conjugates in order to study the impact of the adamantyl group at the 5-position of the thiadiazole ring on both anti-inflammatory activity and COX selectivity. Additionally, the ulcerogenic and cytotoxicity properties of the derivatives were examined to determine their safety profile. The enzymatic activity against COX-1, COX-2 and LOX will be undertaken to identify possible drug actions, and a molecular docking study will be conducted to understand at the molecular level the inhibitory action of the title derivatives.

## 2. Results and Discussion

### 2.1. Chemistry

The title compounds were synthesized following the general method described previously as outlined in Figure 1 [33]. In brief, the starting material 1-adamantyl-5-carbonylchloride was used, which upon reaction with thiosemicarbazide followed by cyclization of the intermediate 1-(1-adamantyl-5-carboxy)thiosemicarbazide in cold sulfuric acid afforded 5-adamantyl-1,3,4-thiadiazole. The latter, upon treatment with chloracetyl chloride, provided 5-adamantyl-1,3,4-thiadiazolyl-2-chloroacetamide, which underwent in heterocyclization reaction by action of ammonium thiocyanate to yield 2-{[5-(adamantan-1-yl)-1,3,4-thiadiazol-2-yl]imino}c-1,3-thiazolidin-4-one. The heating of the previously prepared thiazolidinone with the proper aldehydes furnished the final compounds 5-damantyl-2-(1,3,4-thiadiazole)imino-5-arylidene-4-thiazolidinones. The characterization of compounds is reported in our previous paper [33].

### 2.2. In Vivo Anti-Inflammatory Activity Testing

The assessment of anti-inflammatory efficiency of the prepared derivatives was performed utilizing a known model involving carrageenan-induced mouse paw edema [45,46,47], as the reference drug indomethacin was used. All tested compounds were administrated through i.p. with a molar concentration of 0.028 mmol/kg.

The results of the anti-inflammatory effect of the derivatives using indomethacin as the standard are provided in Table 1 in the form of a percent inhibition of weight gain in the sole muscle pertaining to the right hind leg, by comparison with the left utilized as a reference.

The investigated compounds were shown to induce protection against carrageenan-induced mouse paw edema as outlined in Table 1. The protection ranged from 21.4% to 66.7%, while indomethacin protection was 47% at the same molar concentration. The anti-inflammatory effect of the following derivatives **3**, **4**, **10** and **14** was found to be more potent than indomethacin with compound **4** being the most active in the series, while the derivatives **5**, **7** and **11** exhibited similar effects to that of the control drug. It was observed that the least potent compound was found to be **15** with only 21.4% of protection. The order of activity can be presented as: **4 > 3 > 14 > 10 > 5 > 11 > 7 > 9 > 1 > 2 > 6 = 8 > 13 > 12 > 15**.

The position and nature of substituents on the benzylidene ring of derivatives have deeply impacted the anti-inflammatory activity. In general, the introduction of electron-withdrawing substituents on benzylidene moiety was more favorable for the activity. For instance, when comparing compound **4** (4-NO_2_) with the strongest electron-withdrawing group and compound **6** (4-OCH_3_) with the strongest donating group, derivative **4** with the nitro group at the 4-position was found to be twice more potent than compound **6** (4-OCH_3_). A similar trend was observed with compound **15**, as it was the least active with hydroxy and methoxy substituents at positions 4 and 3, respectively. Derivative **3** with a bromine atom at the 4-position displayed similar potency to that of 4-nitro derivative **4**, while the presence of fluorine at the same position was found to be less favorable to activity, with only 39.7% of protection, which was similar to the unsubstituted compound **1**. The presence of a nitro group at the meta and ortho positions was less beneficial to activity in comparison with para position. As for di-substituted compounds with halogen, in the chlorine series, compound **14** was the most potent with chlorine in the 2 and 6 positions, while derivative **11** with fluorine at the 2 and 6 positions showed lower activity than **14**. However, derivative **10** with fluorine at the 2 and 3 positions demonstrated a favorable activity of 55.8% edema inhibition. According to the structure–activity relationship, the electronic effect of the substituent played a key role in the anti-inflammatory activity of these compounds, where the electron-withdrawing group was found to be the most favorable.

### 2.3. Ulcerogenic Activity

Encouraged by the prominent anti-inflammatory results, the ulcerogenic potency of the selected derivatives **3**, **4** and **14** and the reference drug indomethacin was examined in order to determine if our compounds have better gastric profile than that of indomethacin. Gastric toxicity was conducted through oral administration of 0.028 mmol/kg of the tested derivatives on mice, and the extent of lesions of the gastric mucosa was measured. The ulcerogenic effect of the tested compounds **3**, **4** and **14** revealed that they were devoid of gastric toxicity while the reference drug demonstrated gastric toxicity for four out of five tested mice with an ulcer index of 0.6 (Table 2). This study clearly indicated the superiority of our compounds over indomethacin concerning ulcerogenic liability.

### 2.4. Molecular Target Identification

On the basis of their anti-inflammatory activity, the mechanism of action was prospected against COX-1, COX-2 and LOX for the following selected compounds, **3**, **4** and **14**. The enzymatic assays, presented in Table 3, clearly showed that our derivatives displayed significantly inhibition against the COX-1 enzyme with a micromolar range of IC_50_, and no action against COX-2 was observed. Our derivatives appeared to be more efficient COX-1 inhibitors than both standard drugs, naproxen and ibuprofen, which are more effective COX-1 than COX-2 inhibitors. Moreover, derivatives **3**, **4** and **14** exhibited a modest enzymatic inhibition against LOX. 

In light of the ulcerogenic and enzymatic study, we have demonstrated that the title compounds were identified as a novel class of selective COX-1 inhibitors without causing any sign of gastric damage, which is in agreement with the findings of Tanaka et al. [9] and Wallace et al. [10].

Unfortunately, the anti-inflammatory activity of these compounds cannot be accounted for in the inhibition of COX-1/LOX enzymes. Other mechanisms are likely to be involved in the anti-inflammatory activity of our compounds. The structural change by introducing the adamantyl ring on the scaffold, as we hypothesized, failed to shift COX selectivity in favor of COX-2. 

### 2.5. Docking Studies

Molecular docking was undertaken against the following molecular targets COX-1, COX-2 and 5-LOX in order to rationalize the inhibition behavior of the title derivatives, and the studies are illustrated in Table 4.

According to molecular modeling analysis of the co-crystallized ligand, ibuprofen is anchored in the COX-1 active site, making three hydrogen bonding interactions between its carboxylate group and with residue Tyr355 (O···H, 1.85 Å) and two with residue Arg120 (NH···O, 1.83 Å and NH···O, 1.91 Å) (Figure 3). Furthermore, several hydrophobic contacts with residues Ile523, Val116, Leu351, Leu359, Ala527, Val349, Trp387, Tyr385 and Phe518 were involved. The predicted binding mode of compounds **3**, **4** and **14** (Figure 3, Table 4) showed that the benzylidenethiazolidinone moiety of the derivatives deeply occupied the hydrophobic region of the COX-1 binding site in a similar manner to that of ibuprofen, in which the benlylidene ring was pointed toward the residues Phe381, Tyr385 and Trp387, while the adamantyl moiety was found at the entrance of the active site interacting mainly with Val116, Val119 and Ile89. Similar to ibuprofen, hydrogen bonding interaction was predicted between one of the nitrogen atoms of thiadiazole and the OH residue of Tyr355 (N···H, 1.70 Å) only for compounds **3** and **4**. These key interactions and binding mode can probably explain their good IC_50_ values (1.08 and 1.12 μM) in comparison to compound **14** (9.62 μM). Another key interaction that was predicted is the involvement of pi-cationic interaction between residue Arg120 and the thiadiazole ring observed for all compounds. As for compound **4**, hydrogen bonding and pi-pi interactions were predicted between the nitro group and the amino acid Tyr385, while compound **3** was engaged in several hydrophobic contacts with residues Tyr385, Trp387, Phe381 and Leu384 through its bromine atom. Compound **14**, lacking hydrogen bonding interaction with residue Tyr355, formed pi-sulfur and T-shape interactions between Tyr355 and the thiadiazole ring, which may explain its moderate COX-1 inhibitory activity that is consistent with its lower predicted binding energy (−7.60 kcal/mol). All docked complexes bind in a similar manner as ibuprofen, which is mirrored by their reduced protein–ligand binding and by their elevated inhibitory behavior. On the basis of our predicted docking model, the selectivity of COX-1 over COX-2 could be explained by the fact that compounds **3**, **4** and **14** failed to occupy the hydrophilic region formed by residues Phe518, Ala516, Arg 513, His90, Ser 530 and Gln192 in which selective COX-2 inhibitors are well documented to be strongly involved through hydrogen bonding interaction with their sulfonamido or sulfonyl group. This is the reason why our compounds turned out to be COX-1 selective inhibitors by lacking such hydrophilic pocket occupation crucial for COX-2 activity. Our molecular modeling results were therefore in agreement with the observed COX-1 enzymatic activity of the title compounds.

To further investigate the structural insight into COX-1 selectivity, we conducted a docking study of the tested compound **3** against the COX-2 enzyme active site. In that regard, two COX-2 co-crystal structures were selected. The first one is the co-crystal structure with the selective COX-2 inhibitor celecoxib (PDB 3LN1), and the second is the co-crystal structure with the selective COX-2 inhibitor indomethacin-butyldiamine-dansyl conjugate (PDB 6BL3). The binding domain of both COX-2 receptors differs slightly by the conformation of residue Arg120 (Figure 4). Residue Arg 120 (green) from the celecoxib active site is pointed toward Tyr355, while Arg120 (purple) from the indomethacin conjugate active site is oriented away from Tyr355. 

The binding energies predicted for compound **3** against the cavity site of the co-crystal structures PDB 3LN1 and PDB 6BL3 were found to be −4.11 and −7.36 kcal/mol, respectively, indicating a much less favorable binding interaction for the celecoxib binding site. In the celecoxib active site, compound **3** was predicted to occupy mainly the hydrophilic pocket of the COX-2 active site by making only hydrophobic contacts with the residues Gln 178 and Ala502, and the adamantyl moiety was found outside of the active site, playing any role in COX-2 activity. Although two hydrogen bonding interactions were predicted with residues Arg499 and Tyr341, the hydrophobic COX-2 site was not occupied. In addition, its predicted binding energy (−4.11 kcal/mol) in comparison to that of celecoxib (−9.63 kcal/mol) demonstrated a strongly unfavorable binding interaction. As for the indomethacin conjugate active site (PDB 6BL3), the benzylidene ring failed to deeply occupy the hydrophobic pocket, and the adamantylthiadiazole moiety was located at the entrance of the COX-2 active site while only the adamantyl core was positioned at the entrance of the COX-1 binding site. This indicated that the larger active site of COX-2 allowed for more flexibility and was hence less tightly bound to the receptor than that of COX-1. In addition, no hydrogen binding interaction was observed with the COX-2 target. The difference of predicted binding energy between derivative **3** and those of the celecoxib and indomethacin conjugates indicated that our compound is predicted to be more likely inactive against COX-2. The molecular modeling results are consistent with the observed inhibitory activity since the compounds showed no activity against the COX-2 enzyme. Another observation from this study is that Arg120 was not involved in the binding interactions in both COX-2 active sites.

On the basis of our molecular modeling analysis against COX-1 and COX-2 targets, the COX-1 inhibitory action of our compounds accounted for the involvement of residue Arg120 through a pi-cation interaction, which provides the structural basis for COX-1 selectivity. 

Docking to the 5-LOX active site of the most active compounds revealed that residue Ile367 is a common NDGA binding site for derivative **14**. This is crucial since residue Ile367, together with residues His372, His367 and Arg596, hold a catalytic role in the 5-LOX active site, withholding iron or forming the α2-helix [47]. 

NDGA, utilized as a standard drug in the in vitro test, as is illustrated in Figure 5, was stabilized by four hydrogen bonds with the residual amino acids Ile673 and Gln363 of the 5-LOX active site through its phenolic-hydroxyl carriers. Moreover, NDGA interacts via its oxygen atom of the hydroxyl group with the Fe of the catalytic site of the enzyme. This interaction is also observed for the most active compounds tested: **3**, **4** and **14** (Table 4). Moreover, in compound **14**, the formation of hydrophobic interactions of compound **14** connecting with residues including Ile406, Leu368, Ala410 and Leu414 and the presence of a halogen bond between the chloro substituent of the compound and residue Gln363 apparently increases the enzyme–inhibitor complex stability (Figure 6B). All these interactions stabilize the enzyme–inhibitor complex and justify its inhibitory potency (energy of binding −9.94 kcal/mol, 34.5 ± 1.6% in 100 µM). 

### 2.6. Cytotoxicity Assessment

The evaluation of cellular cytotoxicity of derivatives **3**, **4**, **10**, **11** and **14** in normal human MRC-5 cells was accomplished at three different concentrations in cell culture, i.e., 0.1, 1 and 10 μM (Figure 7). No substantial effect on cell proliferation after 48 h of exposure was found in cultures, as the growth was ≥98% for all assessed agents by comparison with the control (untreated cells) (Figure 7). 

## 3. Materials and Methods

κ-Carrageenan and lipoxygenase type I-B from soybean were obtained from Sigma (St. Louis, MO, USA). COX inhibition was studied utilizing the kit of “COX Inhibitor Screening Assay” (Cayman Chemical Co., Ann Arbor, MI, USA). In vivo tests, male and female (23–30 g) mice R”’ were stored in the Centre of the School of Veterinary Medicine (EL54-BIO42), (Aristotle University of Thessaloniki, Thessaloniki, Greek), licensed by the state official veterinary authorities (presidential degree 56/2013, in standardization with the European Directive 2010/63/EEC). The experimental protocols were approved by the Committee of Animal Ethics of the Prefecture of Central Macedonia (no. 270079/2500).

### 3.1. Chemistry

Compound **10** was synthesized analogous to compounds described in our previous paper [33]. 

2-{[5-(Adamantan-1-yl)-1,3,4-thiadiazol-2-yl]-imino}-5-(2,3-difluorobenzylidene)-1,3-thiazolidin-4-one **10**. Yield, 60%. Rf = 0.56 (petroleum ether: ethyl acetate = 1:1). ^1^H NMR (500 MHz, DMSO-*d*_6_) δ 10.15 (s, 1H, NH), 7.60 (s, 4H), 2.03 (s, 1H), 1.97 (q, *J* = 8.9, 10.6 Hz, 6H), 1.88 (d, *J* = 2.8 Hz, 1H), 1.72 (q, *J* = 3.9, 4.6 Hz, 7H).^13^C NMR (75 MHz, DMSO-*d*_6_) δ 190.75, 180.63, 177.27, 160.21, 136.27, 136.11, 130.25, 130.09, 129.23, 129.07, 127.76, 115.54, 40.46, 38.71 (3C), 36.01 (3C), 31.81, 30.36 (2C). Anal.Calcd. For C_22_H_20_F_2_N_4_O_S2_ (%). C, 57.62; H, 4.40; N, 12.22. Found: C, 57.58; H, 4.47; N, 12.19.

### 3.2. Effect on Mouse Paw Edema in Carrageenan-Induced

The animals were weighed and then randomly divided into control, standard and test groups with each group including 10 mice. Saline intraperitoneal (0.1 mL, control) was administered to the first group of rats, while a dose of 10 mg/kg of the suspension of the standard (indomethacin) was given to the second group, and the tests groups were subjected to equimolar doses of the suspension of the assessed derivatives relative to standard drug. Edema was induced by injecting 0.1 mL of 1% aqueous solution of carrageenan subcutaneously to the sub-plantar region of the right hind paw, utilizing the left paw as control. The assessed derivatives, in suspended water/Tween-80, were injected i.p. (0.15 mmol/kg) 5 min before carrageenan administration. After 3.5 h, the hind paws were excised and were weighed separately. The edema paws were estimated as paw weight increase [48].

### 3.3. Ulcerogenicity Activity

The ulcerogenicity effect of compounds was assessed on male mice by following the reported protocol [49]. Male mice weighing 22–25 g were separated into six groups (five animals in each group). The standard drug indomethacin (0.028 mmol/kg) and the assessed derivatives (0.028 mmol/kg) were suspended in saline solution with the aid of a few drops of Tween^®^80 (Sigma Aldrich, St. Louis, MO, USA) and were orally given for 3 successive days (single dose/day) to the fasted animals. The placebo control group was administrated saline with few drops of Tween-80. The animals were sacrificed 4 h after the last dose, and the stomachs were inspected for stomach lesions. The scoring of ulcerogenicity was expressed in terms of: average number of ulcers per stomach, percentage incidence of ulcers and percent incidence of ulcer divided by 10. The ulcer index was calculated using the sum of the three above values.

### 3.4. COX-1 and COX-2 Activity Inhibition

The action of the tested derivatives on COX-1 and COX-2 inhibition was analyzed utilizing a commercial kit furnished by Cayman (Cayman Chemical Co., Ann Arbor, MI, USA) and by applying the instructions of the manufacturer. The kit utilizes human recombinant COX-2 and ovine COX-1 enzymes. This method is an excellent tool that can be utilized for general inhibitor screening, or to eradicate false positive leads induced by less specific tests. The assay analyzes PGF_2a_ produced by SnCl_2_ reduction of COX-derived PGH_2_. The prostanoid product was quantified using enzyme immunoassay via a broadly specific antibody that binds to all major prostaglandin compounds [48].

### 3.5. LOX Activity Inhibition

The reaction mixture contained the final concentration (300 µΜ) of the test derivatives, dissolved in absolute ethanol (10–300 µM), or the solvent (control), soybean LOX, dissolved in 0.9% NaCl solution (250 u/mL) and sodium linoleate (100 µM), in Tris–HCl buffer, pH 9.0. The reaction was monitored for 7 min at a temperature of 28 °C, noting the absorbance (234 nm) of a conjugated diene structure, following the formation of 13-hydroperoxy-linoleic acid. The performance of the assay was validated utilizing NDGA as a standard. For the evaluation of the inhibition type, the above experiments were repeated, utilizing sodium linoleate (1 mM), which is greater than the saturating substrate concentration [48].

### 3.6. Docking Studies

The molecular modeling simulation was conducted with Molecular Operating Environment (MOE) program using the following X-ray crystal structures COX-1 (PDB code: 1EQG), COX-2 (PDBs: 3LN1 and 6BL3). The force field selected was the Merck molecular force field 94x (MMFF94x). The triangle method was selected as the placement method. London dG and DBV/WSA dG were used as a first and a second rescoring method, respectively. Induced fit was used for docking process. Finally, the 3D and 2D binding interactions were visualized by using Accelrys Discover Studio software.

Molecular modeling assessments were also accomplished utilizing AutoDock 4.2 [36,50], and the X-ray crystal structure of 5-LO (PDB ID: 6N2W) bound to NDGA was downloaded from Brookhaven Protein Data Bank (PDB). All procedures were performed as to our former studies [47]. 

The molecules were sketched in chemdraw 12.0 program. The geometry of built compounds was optimized utilizing the molecular mechanical force fields 94 (MMFF94) energy with program LigandScout (ver. 4.4.5), partial charges were additionally computed, conformers of each ligand were produced, and the one with the best conformation was kept and saved as mol2 files that were passed to ADT for pdbqt file preparation. The polar hydrogen was added to each structure, followed by computing Gasteiger and Kollman charges and torsions. The region of interest, used by Autodock4 for docking runs and by Autogrid4 for affinity grid maps preparation, was defined in such a way to include the entire catalytic binding site via a grid size of 50 × 50 × 50 xyz points with grid spacing of 0.375Å. 

For the docking simulation, translation, quaternation and torsion steps were applied following default values. The default parameters of the Lamarckian Genetic Algorithm were used for minimization. After docking completion (100 runs), examination of binding energy scores (Δ*G*_binding_, kcal/mol) and numbers in cluster lead to the disclosure of the best poses. In order to describe the ligand-binding pocket interactions, the top ranked binding mode found by AutoDock in the complex with the binding pocket of the enzyme was selected. Regarding the graphical representations of all the ligand–protein complexes, LigandScout (ver. 4.4.5) was utilized. 

The co-crystallized ligands of enzymes COX-1, COX-2 and 5-LOX were removed and docked into the active site of the enzymes in order to validate the docking protocol. The studies showed that the docked ligands ibuprofen and NDGA seemed exactly superimposed on the co-crystallized bound ones with a root mean square deviation (RMSD) of 1.12 and 0.85 Å, respectively, indicating the ratability of our docking protocol. 

### 3.7. Cytotoxicity Studies

The MRC-5 human cell line from lung fibroblast was stored and then utilized in our laboratory in a conventional manner (passage < 40). MRC-5 cells were cultured (using a humidified atmosphere enclosing 5% CO_2_ at 37 °C) in medium DMEM with 10% *v*/*v* FBS, 1% PS penicillin–streptomycin. The assessed derivatives were dissolved in *dimethyl sulfoxide* and stored at a temperature of 4 degrees centigrade. For cytotoxicity assessment, cells were seeded in 96-well culture plates at 5 × 10^4^ cells/mL (initial density) and permitted to attach for at least 3 h before adding the compounds at increasing concentrations (0.1, 1 and 10 µM). *Dimethyl sulfoxide* concentration used in the culture did not exceed 0.2% *v*/*v*, showing no detectable effect on cell proliferation [51,52]. To evaluate the cytotoxicity of each derivative, the cells were given an additional 48 h to grow before estimating their number in culture utilizing the Neubauer counting chamber (microscope observation). In every treated culture, cell growth was expressed and reported as the percentage of untreated growing cells. GraphPad Prism 6.0 program was utilized in order to accomplish statistical *t*-test analysis.

## 4. Conclusions

The use of conventional NSAIDs for the treatment of pain and inflammation leads to severe side effects such as gastric ulceration and cardiotoxicity. As a consequence, the discovery of novel anti-inflammatory agents endowed with reducing side effects is of paramount importance. In this context, a series of 5-adamantylthiadiazole-based thiazolidinone derivatives was investigated for their anti-inflammatory activity, mode of action, ulcerigenocity and cytotoxicity. It was observed that derivatives **3**, **4**, **10** and **14** displayed higher anti-inflammatory activity versus that of the control drug indomethacin. Compound **4** with a nitro group in the para position of the benzylidene ring was identified as the most potent agent with 66.7% of edema inhibition, while indomethacin showed 47% edema inhibition. According to our findings, the anti-inflammatory activity of the tested derivatives depends on the substituent and their position in the benzene ring. More preferably, it appeared to be substituents with electron-withdrawing groups. The enzymatic assays of the tested derivatives against COX and LOX revealed significant inhibition against the COX-1 enzyme. Molecular modeling provided the structural basis for COX-1 selectivity in which involvement of the residue Arg120 forming pi-cation interactions was responsible for the selectivity. The ulcerogenic activity of the most potent derivatives, **3**, **4** and **14**, demonstrated no sign of gastric lesions, indicating that our compounds showed superiority over indomethacin with respect to gastric toxicity. In addition, the study of cytotoxicity of the derivatives conducted against normal human MRC-5 cell line showed no sign of toxicity. Therefore, 5-adamantylthiadiazole-based thiazolidinone conjugates were identified as a novel class of selective COX-1 inhibitors with good gastric and toxicity profiles, displaying promising anti-inflammatory activity. Unfortunately, the anti-inflammatory activity of these compounds could not be attributed to COX-1/LOX inhibition, and further work is needed to clarify their drug action. However, the 5-adamantylthiadiazole-based thiazolidinones could be used as a template to explore other COX-1-related disorder diseases.

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
