# Peer review of "Identification of Novel Cyclooxygenase-1 Selective Inhibitors of Thiadiazole-Based Scaffold as Potent Anti-Inflammatory Agents with Safety Gastric and Cytotoxic Profile"

_molecules, 2023, doi:10.3390/molecules28083416_

Round 1

Reviewer 1 Report

Manuscript entitled, “Identification of novel cyclooxygenase-1 selective inhibitors of 2 thiadiazole-based scaffold as potent anti-inflammatory agents 3 with safety gastric and cytotoxic profile.

1. The abstract section should be more informative; authors are advised to reorganize with brief information regarding the problem and add significant values of your results as well add the value for indomethacin.

2. Are these molecules are selective and why, any reason?

3. Are all these compounds are new or first time reported for their reported biological activities?

4. Authors are advised to improve English; several grammatical errors are there in the manuscript.

5. In Cytotoxicity assessment why you selected 3 concentrations with highest 10 µM.

6. The docking score is almost very close is there any reason?

7. Conclusion is very short make it more elaborative and informative.

8. Overall, the manuscript is very well written, and the information’s are clear and is recommended for acceptance.

Author Response

First of all, we would like to thank the referee for accepting in reviewing our manuscript. Kindly be informed that both abstract and conclusion have been updated and a paragraph in molecular modelling has been added in an attempt to rationalize the cox selectivity of our compounds. 

Manuscript entitled, “Identification of novel cyclooxygenase-1 selective inhibitors of 2 thiadiazole-based scaffold as potent anti-inflammatory agents 3 with safety gastric and cytotoxic profile.

  1. The abstract section should be more informative; authors are advised to reorganize with brief information regarding the problem and add significant values of your results as well add the value for indomethacin.

Answer:  The abstract has been updated according to your suggestions

  1. Are these molecules are selective and why, any reason?

Answer:  On the basis of the enzymatic assay, our compounds inhibit only COX-1 enzyme However, we have attempted to rationalize the cox-1 selectivity by docking studies. Two paragraphs have been added in page 8 and 9 of the manuscript.

  1. Are all these compounds are new or first time reported for their reported biological activities?

Answer: These compounds were reported previously as antimicrobial agents and now we report their anti-inflammatory activity. However, Compound 10 was never reported and we have included the characterization details in page 12.

  1. Authors are advised to improve English; several grammatical errors are there in the manuscript.

Answer: We took into account your suggestions and improved the language.

  1. In Cytotoxicity assessment why you selected 3 concentrations with highest 10 µM.

Answer: Regarding this point raised by the reviewer that deals with the issue of the selected logarithmic concentration range 0.1, 1, and 10 μΜ in the experiment of cytotoxicity assessment, we must say that this is part of the standard pharmacological procedure upon attempting to evaluate the potential IC50 value of the agents under investigation. In the case, that cytotoxicity in the cell cultures is measured then the concentration range is enlarged covering numbers beyond this window in order the estimated IC50 value to be accurate. If, however, no cytotoxicity is observed within this initial logarithmic concentration range (0.1, 1, and 10 μΜ), and this is the case in our data presented in the manuscript, then no additional experimentation range is needed to further assess the cytotoxicity of the compound under investigation. Considering the highest 10 μM value, we must emphatically say that compounds showing effect higher than 10 μM, e.g., within the range of 100 μΜ in cell cultures, are considered no promising, since in the opposite case they will need to be administered in the organism in the range of grams to exert their observed pharmacological effect, something than does not coincide with the drug development process. Actually, by having pharmacological effects at the cellular level within the range of concentrations >10 μM this implies no pharmacologically active compound in order to follow the next steps for drug development. The latter, of course, can be done by structurally modifying the compound in order the new agents to achieve an active pharmacological profile at <10 μΜ concentrations.

  1. The docking score is almost very close is there any reason?

Answer: These three compounds that were studied by docking were the best in terms of docking score compared to other compounds which had lower docking score from -4.63 to -7.11

  1. Conclusion is very short make it more elaborative and informative.

Answer: Conclusion has been updated.

  1. Overall, the manuscript is very well written, and the information’s are clear and is recommended for acceptance.

Answer: Thank you for your positive decision.

Reviewer 2 Report

In this MS, authors have presented a series of compounds claiming their anti-inflammatory activity due to their targeting COX-1. Their are several scientific issues for which a common reader need justification. The authors need to address these issues before publishing the MS. 

1. Role of COX-1 in cytoprotective effect on gastric functions is well documented. How do authors claim their compounds are COX-1 inhibitors but do not cause gastric toxicity? 

2. even if inhibition of both COX-1 and 2 causes gastric toxicity, the compounds in line are inhibiting COX1 but nil ulcerogenic activity (Table 2); how do authors justify it?

3. There is no rationale for the design of molecules. Similar molecules are reported as COX2 inhibitors as well...what leads to the selectivity of these compounds for COX1?

4. What is the role of adamantyl unit? Size of COX-2 active site is larger than that of COX1...presence of adamantyl unit may make the compounds more of COX-2 selective.

5.  experimental proof for the anti-inflammatory activity of the compounds due to COX-1 inhibition is not convincing.

6. As per the data in table 3, compds are showing 100% enzyme inhibition at 20 uM while IC50 are calculated 1/2 uM?

      in the light of these issues, this reviewer is of the opinion that the MS in its present is not suitable for publicaion.

Author Response

First of all, we would like to thank the referee for accepting in reviewing our manuscript and providing constructive remarks for manuscript improvement. We hope that all points raised by the referee have been successfully addressed.

In this MS, authors have presented a series of compounds claiming their anti-inflammatory activity due to their targeting COX-1. They are several scientific issues for which a common reader need justification. The authors need to address these issues before publishing the MS. 

  1. Role of COX-1 in cytoprotective effect on gastric functions is well documented. How do authors claim their compounds are COX-1 inhibitors but do not cause gastric toxicity? 

Answer: We agreed with the role of COX-1 in cytoprotective effect on gastric functions. However, numerous studies demonstrated that selective COX-1 inhibitors with anti-inflammatory activity are devoid of gastric toxicity and these findings could be contrary to present knowledge. Our introduction part was devoted to that concerns in which selective cox-1 inhibitors do not display any sign of gastric damages (see ref: 9, 10, 11, 12, 13, J. Med. Chem. 2006, 49, 7774-7780 (Synthesis of 2-Methyl-3-indolylacetic Derivatives as Anti-Inflammatory Agents That Inhibit Preferentially Cyclooxygenase 1 without Gastric Damage) and Bioorganic & Medicinal Chemistry 25, 2017, 665–676 (Design, synthesis and analgesic/anti-inflammatory evaluation of novel diarylthiazole and diarylimidazole derivatives towards selective COX-1 inhibitors with better gastric profile).  In addition, molecules presented in figure 1 have these characteristics as well (please see ref 15 and 16)

The study of ulcerogenic effects of our compounds was originally initiated from a reviewer of our previous submitted manuscript entitled “Discovery of 5-Methylthiazole-Thiazolidinone Conjugates as Potential Anti-Inflammatory Agents: Molecular Target Identification and In Silico Studies” (Molecules 202227(23), 8137; https://doi.org/10.3390/molecules27238137) requested us to perform the ulcerogenic effects of our compounds to demonstrate the superiority gastric profile of our compounds compared to indomethacin.

In the current MS, the ulcerogenic effects have been conducted with the same molar concentration to that of anti-inflammatory activity assessment with respect to indomethacin (10 mg/Kg). At this concentration, we did not observe any sign of GI toxicity but indomethacin did.

  1. even if inhibition of both COX-1 and 2 causes gastric toxicity, the compounds in line are inhibiting COX1 but nil ulcerogenic activity (Table 2); how do authors justify it?

Answer: As we mentioned in introduction, Tanaka et al., demonstrated that cyclooxygenase-1 selective inhibition up-regulates cyclooxygenase-2 expression in gastric mucosa resulting in an increase of intestinal prostaglandin E2 synthesis in order to compensate the decrease of prostaglandin E2 level through COX-1 inhibition and hence the COX-2 up-regulation is responsible for the maintenance of the mucosal integrity (ref 14). This is the reason why COX-1 selective inhibitors are safe for GI tract and this is what we have observed in our experimental results.

  1. There is no rationale for the design of molecules. Similar molecules are reported as COX-2 inhibitors as well...what leads to the selectivity of these compounds for COX1?

Answer: Please note that originally, we thought to replace the methyl group in the 5-Methylthiazole-Thiazolidinone series by a bulky group aiming at reverting the COX selectivity towards COX-2, but surprisingly our enzymatic assay results proved the contrary. This rationale has been added in MS.

In addition, we have provided an explanation to rationalize the cox-1 selectivity through docking studies. Two paragraphs have been added in page 8 and 9 of MS.

  1. What is the role of adamantyl unit? Size of COX-2 active site is larger than that of COX1...presence of adamantyl unit may make the compounds more of COX-2 selective.

Answer: Please note that we have addressed this concern by reporting the docking study against COX-2 in molecular modelling part (page 9) and we have also provided additional explanation on the selectivity in page 8.

  1. Experimental proof for the anti-inflammatory activity of the compounds due to COX-1 inhibition is not convincing.

Answer: Please note that we agree that COX-1 is not the molecular target for their anti-inflammatory activity but our research group worked mainly on COX-1, COX-2 and LOX molecular targets, this is the reason why we have conducted the enzymatic assay against these three targets and reported them. However, we have added a paragraph in page 7 (highlighted in yellow) to address this issue. Kindly note that the abstract, introduction and conclusion have been updated accordingly.

  1. As per the data in table 3, compds are showing 100% enzyme inhibition at 20 uM, while IC50 are calculated 1/2 uM?

Answer: Please note that the inhibition concentration for COX was conducted at 200 uM not at 20 uM, as a consequence the measured IC50 are much more relevant.

Reviewer 3 Report

The manuscript is devoted to design of novel non-steroidal anti-inflammatory agents not causing such severe side-effect as gastric ulcers. Authors studied a series of 5-adamantylthiadiazole-based thiazolidines, they found a compound surpassing indomethacin in inhibition of COX-1 enzyme. The main advantage of studied compounds is good gastric and toxicity profile.

The article is suitable for publication after revision.

As a comment, the main point is Chemistry chapter. All compounds except 2,3-difluorophenyl derivative were reported in the article [Biomol Med Chem 2018, v, 26, p. 4664], spectral data for compound 10 are absent in the experimental part of the current manuscript. Moreover, the conditions presented in the Scheme 1 are not the same as in earlier article. Although including aqueous ammonia as alkalizing agent is really helpful, the temperature at the stage a is not the same (37 deg instead of 25 deg), at stage d keeping at room temperature overnight is missing, the duration at the stage e is not exactly the same (3-5 h in the current manuscript and 4 h in earlier one). Differences and may be some improvements are not discussed in the paragraph (lines 126-136).

Author Response

The manuscript is devoted to design of novel non-steroidal anti-inflammatory agents not causing such severe side-effect as gastric ulcers. Authors studied a series of 5-adamantylthiadiazole-based thiazolidines, they found a compound surpassing indomethacin in inhibition of COX-1 enzyme. The main advantage of studied compounds is good gastric and toxicity profile.

The article is suitable for publication after revision.

As a comment, the main point is Chemistry chapter. All compounds except 2,3-difluorophenyl derivative were reported in the article [Biomol Med Chem 2018, v, 26, p. 4664], spectral data for compound 10 are absent in the experimental part of the current manuscript. Moreover, the conditions presented in the Scheme 1 are not the same as in earlier article. Although including aqueous ammonia as alkalizing agent is really helpful, the temperature at the stage a is not the same (37 deg instead of 25 deg), at stage d keeping at room temperature overnight is missing, the duration at the stage e is not exactly the same (3-5 h in the current manuscript and 4 h in earlier one). Differences and may be some improvements are not discussed in the paragraph (lines 126-136).

Answer:

First of all, we would like to thank the referee for accepting in reviewing our manuscript. Kindly note that both abstract and conclusion have been updated and a paragraph in molecular modelling has been added in an attempt to explain the cox selectivity. 

Indeed, almost all compounds were published in Bioorg.Med.Chem 2018 as antimicrobial agents. We really appreciated to notice that 2,3-di-F derivative has not yet been reported. We have included the spectral data of compound 10 in experimental part (page 12). You are also right regarding reaction’s conditions in Scheme 1. We apologize for this gaffe. We corrected.

Round 2

Reviewer 1 Report

Author addressed my comments

Author Response

Dear reviewer

Thank very much you for reviewing our manuscript and for providing your acceptance.

Please accept our best regards

Christophe Tratrat 

Reviewer 2 Report

The revised version of the MS was read with interest. Although some strong claims are made, the MS suffers from certain basic scientific flaws.

     It appears that the development of COX-1 inhibitors as anti-inflammatory agents with safe gastric profile in this MS is based on the information reported in Ref 9-12 (as cited by the authors). 

1.  "However, the GI toxicity of traditional NSAIDs is considered to be induced by non-selective inhibition of both COX-2 and COX-1 [9]. Indeed, the inhibition of both COX-1 and COX-2 at the same time was responsible for the gastric ulceration [9]." It is obvious that the NSAIDs cause gastric ulceration due to COX-1 inhibition and the same is advocated by the authors of ref 9. It should be taken in a way that gastric ulceration is caused when both COX-1 and COX-2 are inhibited. 

2. Similar information (as mentioned in comment 1 above) is given in the paper of Ref 10.

3. As per the information available in Ref 11, inhibition of COX-1 upregulates COX-2 expression while these studies are silent about the effect of COX-2 expression in inducing inflammation. It is obvious that inhibition of COX-1 shunts arachidonic acid metabolism to COX-2 channel that may increase the production of inflammatory PGs. Moreover, the authors of ref 11 concluded that more studies are needed to prove their hypothesis (they focused only on gastric damage). 

4. Mofezolac, a selective COX-1 inhibitor also suffers from gastric bleeding. 

5. For the design strategy, Authors state "to revert the COX selectivity in favor of COX-2..." but they are developing COX-1 inhibitors. Moreover, size of COX-2 active site is larger than that of COX-1, how the presence of adamantyl (in place of their previous molecules with CH3 group) will make the molecule more COX-1 selective?

6. It is difficult to understand the efficacy of the compounds from data in Table 1. Instead of percentage inhibition in Table 1, it should be in terms of IC50. Rather than saying 10 mg/kg, authors should mention the molar concentrations of the test compounds. 

7. The toxicity of the compounds on animals must be checked. 

8. There is no mention of what concentrations of the compounds were used for COX-/2 assays; then how IC50 values were calculated? As per data in Table 3, compound concentration used is 200 uM and COX-1 inhibition is 100%...from this data how IC50 comes out be ~1 uM?

9. Authors must check the binding constants of the compounds with COX-1, COX-2 and LOX.

10. Mechanistic studies are needed to prove the proposed COX-1 selective inhibition. 

11. It is not clear what authors want to demonstrate with Figure 4. Why COX-1, COX-2 active sites are superimposed? It is difficult to visualize the selectivity of the compound for COX-1 from the given docking figures. It may be better understandable if 3D views or the surface views of the binding pockets are given. Otherwise, it is not clear which part of Y355 is interacting with compound 4 in Fig 3B. Do authors observed any role of adamantyl moiety from the docking studies because in contrast to their statement, no interaction of this moiety with Val116 is visible in Fig 3B. 

12. Remember, a single H-bonding of the compound with the enzyme (as it is happening for compound 3 with COX-2, Table 4) should result into the some inhibition of the enzyme activity. However, data of Table 3 shows no inhibition of COX-2. 

13. The molecules have PAS assay interference fragments. Did the authors check promiscuous nature of the compounds?

      Therefore, in the light of above reservations, it is not justifiable to publish this data. By taking note of these issues, authors may perform more experiments to prove their hypothesis. 

Author Response

We would like to thank the reviewer once again for using its precious time to provide additional feedbacks, thus helping us in improving the MS. We sincerely hope that the new changes made in the revised MS would be perceived positively by the reviewer.

The revised version of the MS was read with interest. Although some strong claims are made, the MS suffers from certain basic scientific flaws.

     It appears that the development of COX-1 inhibitors as anti-inflammatory agents with safe gastric profile in this MS is based on the information reported in Ref 9-12 (as cited by the authors). 

  1. "However, the GI toxicity of traditional NSAIDs is considered to be induced by non-selective inhibition of both COX-2 and COX-1 [9]. Indeed, the inhibition of both COX-1 and COX-2 at the same time was responsible for the gastric ulceration [9]." It is obvious that the NSAIDs cause gastric ulceration due to COX-1 inhibition and the same is advocated by the authors of ref 9. It should be taken in a way that gastric ulceration is caused when both COX-1 and COX-2 are inhibited. 

Answer: Thank you for the judicious comment. Kindly note that this part of the introduction has been updated with the right corresponding references. Please note that reference 9 (Borer, J.S.; Simon, L.S. Cardiovascular and gastrointestinal effects of COX-2 inhibitors and NSAIDs: achieving a balance. Arthritis Res. Ther., 2005, 7, S14, doi:10.1186/ar1794) was removed. The new reference 9 becomes: Tanaka, A.; Araki, H.; Komoike, Y.; Hase, S.; Takeuchi, K. Inhibition of both COX-1 and COX-2 is required for development of gastric damage in response to nonsteroidal antiinflammatory drugs. Journal of Physiology-Paris, 2001, 95, 21-27.

Please also note that the sentence “the gastric ulceration is caused when both COX-1 and COX-2 are inhibited” has been added to MS.

  1. Similar information (as mentioned in comment 1 above) is given in the paper of Ref 10.

Answer: Indeed, you are right. Please note that this part of the introduction has been updated with the right corresponding references. Thus, the reference 10 has been updated with the right one. The new reference 10 becomes: Wallace, J.L.; McKnight, W.; Reuter, B.K.; Vergnolle, N. NSAID-induced gastric damage in rats: requirement for inhibition of both cyclooxygenase 1 and 2. Gastroenterology, 2000, 119, 706-714.

  1. As per the information available in Ref 11, inhibition of COX-1 upregulates COX-2 expression while these studies are silent about the effect of COX-2 expression in inducing inflammation. It is obvious that inhibition of COX-1 shunts arachidonic acid metabolism to COX-2 channel that may increase the production of inflammatory PGs. Moreover, the authors of ref 11 concluded that more studies are needed to prove their hypothesis (they focused only on gastric damage). 

Answer: You are right for this reference which is presently ref 9. Kindly note that later the same authors (ref 12, Tanaka et al, 2002) have demonstrated the increasing production of PGs by COX-2 when COX-1 was solely inhibited and the exact mechanism of the up-regulation COX-2 expression, when COX-1 is solely inhibited, still remains unknown.  However, we are refraining to address detailed mechanistic pathway events regarding this issue but we have presented mainly the gastric toxicity concern in general way. Kindly note the introduction has been updated.

  1. Mofezolac, a selective COX-1 inhibitor also suffers from gastric bleeding. 

Answer: Unfortunately, we did not find any relevant references for Mofezolac dealing with gastric bleeding. However, we have added its ulcer index and its corresponding article as cited reference (written in Japanese, with table results provided in English), lines 101-102. Moreover, the Figure 1 has been updated with 3 additional COX-1 selective inhibitors with safe gastric profile and text was added accordingly in lines 103-106.

  1. For the design strategy, Authors state "to revert the COX selectivity in favor of COX-2..." but they are developing COX-1 inhibitors. Moreover, size of COX-2 active site is larger than that of COX-1, how the presence of adamantyl (in place of their previous molecules with CH3 group) will make the molecule more COX-1 selective?

Answer: Indeed, we have mentioned reverting COX selectivity due to your previous comment raised in part 3 first round, which was

“There is no rationale for the design of molecules. Similar molecules are reported as COX-2 inhibitors as well...what leads to the selectivity of these compounds for COX1?”

Our reply was “ Please note that originally, we thought to replace the methyl group in the 5-Methylthiazole-Thiazolidinone series by a bulky group aiming at reverting the COX selectivity towards COX-2, but surprisingly our enzymatic assay results proved the contrary. This rationale has been added in MS.”  And we added to the text in introduction “The rationale of the design molecules was to revert the COX selectivity in favor to COX-2 by exploiting the size difference between COX-2/COX-1 active sites. It appeared to us that the adamantyl group would be the ideal substituent to achieve this objective”.

We thought it was convenient to mention our first original plan for testing such compounds based on your comment. However, we are ready to revert the original manuscript focusing on COX-1 selective inhibitors if required.

From your second part “how the presence of adamantyl (in place of their previous molecules with CH3 group) will make the molecule more COX-1 selective”: we have never claimed that the presence of adamantyl group will make more COX-1 selective inhibitor in comparison with methyl since methyl derivatives were already COX-1 selective inhibitors. In addition, the adamantyl and methyl series have similar COX-1 inhibitory activity at micromolar range, but we have observed an improvement in anti-inflammatory activity (see figure below).

  1. It is difficult to understand the efficacy of the compounds from data in Table 1. Instead of percentage inhibition in Table 1, it should be in terms of IC50. Rather than saying 10 mg/kg, authors should mention the molar concentrations of the test compounds. 

Answer: Most of research articles report the anti-inflammatory of tested compounds as % inhibition with a specific concentration and the control drug which is used as reference to determine the efficacy of tested compounds. Below are some examples.

  1. Theodosis-Nobelos, P.; Papagiouvannis, G.; Kourounakis, P.N.; Rekka, E.A. Active anti-inflammatory and hypolipidemic derivatives of lorazepam. Molecules 2019, 24, 3277.
  2. Theodosis-Nobelos, P.; Papagiouvannis, G.; Pantelidou, M.; Kourounakis, P.N.; Athanasekou, C.; Rekka, E.A. Design, synthesis and study of nitrogen monoxide donors as potent hypolipidaemic and anti-inflammatory agents. Molecules 2020, 25, 19.
  3. Theodosis-Nobelos, P., Kourti, M., Gavalas, A., Rekka, E.A.

Amides of non-steroidal anti-inflammatory drugs with thiomorpholine can yield hypolipidemic agents with improved anti-inflammatory activity Bioorg.Med Chem Lett.  2016, 26(3), pp. 910–913.

  1. Kamat, V.; Santosh, R.; Poojary, B.; Nayak, S.P.; Kumar, B.K.; Sankaranarayanan, M.; Faheem Khanapure, S.; Barretto, D.A.; Vootla, S.K. Pyridine- and Thiazole-Based Hydrazides with Promising Anti-inflammatory and Antimicrobial Activities along with Their In Silico ACS Omega. 2020, 5(39), 25228-25239.
  2. Bari, S.B.; Firake, S.D. Exploring Anti-inflammatory Potential of Thiazolidinone Derivatives of Benzenesulfonamide via Synthesis, Molecular Docking and Biological Evaluation. Antiinflamm Antiallergy Agents Med. Chem. 2016, 15(1), 44-53.
  3. A. Abu-Hashem , M.A. Gouda , F.A. Badria.Synthesis of some new yrimido[20,10:2,3]thiazolo[4,5-b]quinoxaline derivatives as anti-inflammatory and analgesic agents. Eur.J.Med.Chem. 45 (2010) 1976-1981.

The molar concentration used was 0.028 mmol/kg of tested compounds, we have added “All tested compounds were administrated through i.p. with a molar concentration of 0.028 mmol/kg” in section 2.2 (results & discussion) lines 175-176.

  1. The toxicity of the compounds on animals must be checked. 

Answer: For the time being, we are afraid of not being able to run such useful experiment. But please note that we have demonstrated from the cytotoxicity studies that our compounds are not carcinogenic.

  1. There is no mention of what concentrations of the compounds were used for COX-1/2 assays; then how IC50 values were calculated? As per data in Table 3, compound concentration used is 200 uM and COX-1 inhibition is 100%...from this data how IC50 comes out be ~1 uM?

Answer: On Table 3, it is mentioned the concentration of the compounds used for the % inhibition study.  The compound concentration used for COX-1/-2 and LOX assay was 200 micromolar and 100 micromolar respectively. To determine the IC50, two independent experiments were conducted: first we identified potential inhibitor at 200 micromolar and then we have determined at each concentration 50, 25, 10, 1 and 0.1 micromolar the %inhibition in order to construct the IC50 curve for IC50 determination.

  1. Authors must check the binding constants of the compounds with COX-1, COX-2 and LOX.

Answer: We are afraid that we will not be able to conduct such experiment in the current time and this study might not provide any additional information on the basis of our enzymatic assay and ulcerogenic assay outcomes. As our compounds demonstrated COX-1 selective and devoid of gastric activity, we cannot see clearly how the binding constant determination with COX-1, COX-2 and LOX will provide additional information.

  1. Mechanistic studies are needed to prove the proposed COX-1 selective inhibition.

Answer: Please note that we have already proved that our compounds are COX-1 selective inhibitors from enzymatic assay and from ulcerogenic assay. They did not cause any gastric toxicity which is in agreement with the study by Wallace et Tanaka for COX-1 selective inhibition. We have performed two independent experiments to prove COX-1 selective inhibition of our compounds. Moreover, the docking results strongly support the observed inhibitory activity in favor of COX-1.

  1. It is not clear what authors want to demonstrate with Figure 4. Why COX-1, COX-2 active sites are superimposed? It is difficult to visualize the selectivity of the compound for COX-1 from the given docking figures. It may be better understandable if 3D views or the surface views of the binding pockets are given. Otherwise, it is not clear which part of Y355 is interacting with compound 4 in Fig 3B. Do authors observed any role of adamantyl moiety from the docking studies because in contrast to their statement, no interaction of this moiety with Val116 is visible in Fig 3B. 

Answer: Kindly be informed that we have undertaken again the docking study on COX-1 and COX-2 using another molecular software MOE in order to provide the structural basis of COX-1 selectivity and to explain the observed enzymatic activity of our compounds. In this study, we have also selected two others COX-2 co-crystal structures with better resolution than the previous one used. The new 3D & 2D interaction figures are provided and the text has been updated accordingly.

In the docking 2D and 3D figures, we can clearly see the involvement of Val116 and others residues with adamantyl core. Y355 is involved in hydrogen bonding donor interaction from its OH and can be clearly seen in 3D interaction figure which can be expanded to see more detailed interactions. However, 2D figure provides only the type of interaction and the corresponding amino acid involved in, but not on which part of amino acid will be involved in H-bonds donor or acceptor, only 3D figure can provide detailed interactions.

  1. Remember, a single H-bonding of the compound with the enzyme (as it is happening for compound 3 with COX-2, Table 4) should result into the some inhibition of the enzyme activity. However, data of Table 3 shows no inhibition of COX-2. 

Answer: Yes, you might be right, but there are many other factors to be considered for activity prediction through computational study such as the strength of hydrogen bonding interaction and the magnitude of predicted binding energy of the investigated compound and solvation which cannot be estimated by docking. If the predicted binding energy of the investigated compound is too low (absolute value) to that of the reference co-crystal ligand and predicts one or several strong H-bond interactions, therefore the compound would be expected to be inactive. In contrary, if the binding energy is high and the predicted binding pose seems to be irrelevant, therefore the compound would be expected to be inactive. However, we have conducted again the docking study with two others COX-2 co-crystal structures with better resolution than the previous one used in order to explain the observed inhibitory action of our compounds.

  1. The molecules have PAS assay interference fragments. Did the authors check promiscuous nature of the compounds?

Answer: Since you have raised this issue, we were interested to check PAINS and promiscuous nature of the most active derivatives. We have used SwissADME and Hit Dexter free webservers to assess PAINS and promiscuous and both are reported in the following table:

Alerts for PAINS

SwissADME

Hit Dexter: Probability and prediction confidence of a compound being moderately or highly promiscuous

PSA (primary screening assays)

Hit Dexter: Probability and prediction confidence of a compound being moderately or highly promiscuous

CDRA (confirmatory dose-response assays)

Most active compounds

moderate or high promiscuous

highly promiscuous

moderate or high promiscuous

highly promiscuous

4-Br

0

0.18

0.27

0.05

0.27

4-NO2

0

0.22

0.29

0.07

0.33

2,6-diCl

0

0.12

0.23

0.04

0.26

2,6-diF

0

0.07

0.22

0.06

0.22

2,3-diF

0

0.09

0.19

0.05

0.21

4-CH3

0

0.2

0.35

0.27

0.42

On the basis of PAINS alert and Hit Dexter predictive tools, our compounds are predicted to have no alert PAINS and display low hit rates in PSA and CDRA and are hence regarded as potentially low promiscuous compounds with the exception of 4-CH3 derivative.

In addition, the cytotoxicity study of our compounds proved the low promiscuous since they found safe against normal cell

  1. Therefore, in the light of above reservations, it is not justifiable to publish this data. By taking note of these issues, authors may perform more experiments to prove their hypothesis. 

Answer: First of all, kindly note that we have identified a new scaffold as selective COX-1 inhibitor. As mentioned in introduction, only few COX-1 selective inhibitors were reported so far as compared to COX-2 inhibitors because it was considered for many years that COX-1 was responsible for gastric toxicity hampering the development of new COX-1 drugs. Nowadays, COX-1, as molecular target, regained increasing interest in COX-1 related disorder diseases. As such, our research work truly deserves to be published.

Secondly, please consider all the relevant additions (in red) that contribute to the improvement of the manuscript.

Finally, please note that we have conducted numerous experiments such as anti-inflammatory, enzymatic essays (COX-1, COX-2, LOX), ulcerogenic and cytotoxicity activities, in addition to updating docking studies to provide the rationale into COX-1 selectivity that makes our manuscript rich in findings.
